# Seeing is Believing? Mitigating OCR Hallucinations in Multimodal Large Language Models

**Zhentao He**[1,†]   **Can Zhang**[1,†]   **Ziheng Wu**[1,✉]   **Zhenghao Chen**[1]
**Yufei Zhan**[1,2]   **Yifan Li**[1,3]   **Zhao Zhang**[1]   **Xian Wang**[1]   **Minghui Qiu**[1,✉]

[1]ByteDance   [2]CASIA   [3]RUC

## Abstract

Recent advancements in multimodal large language models (MLLMs) have enhanced document understanding by integrating textual and visual information. However, existing models exhibit incompleteness within their paradigm in real-world scenarios, particularly under visual degradation (*e.g.*, blur, occlusion, low contrast). In such conditions, the current response paradigm often fails to adequately perceive visual degradation and ambiguity, leading to overreliance on linguistic priors or misaligned visual-textual reasoning. This difficulty in recognizing uncertainty frequently results in the generation of hallucinatory content, especially when a precise answer is not feasible. To better demonstrate and analyze this phenomenon and problem, we propose KIE-HVQA, the first benchmark dedicated to evaluating OCR hallucination in degraded document understanding. This dataset includes test samples spanning identity cards, invoices, and prescriptions, with simulated real-world degradations and pixel-level annotations for OCR reliability. This setup allows for evaluating models' capacity, under degraded input, to distinguish reliable visual information and answer accordingly, thereby highlighting the challenge of avoiding hallucination on uncertain data. To achieve vision-faithful reasoning and thereby avoid the aforementioned issues, we further introduce a Group Relative Policy Optimization (GRPO)-based framework featuring a novel reward mechanism. By incorporating a self-awareness of visual uncertainty and an analysis method that initiates refusal to answer to increase task difficulty within our supervised fine-tuning and reinforcement learning framework, we successfully mitigated hallucinations in ambiguous regions. Experiments on Qwen2.5-VL demonstrate that our 7B-parameter model achieves a ∼28% absolute improvement in hallucination-free accuracy over GPT-4o on KIE-HVQA and there is no significant performance drop in standard tasks, highlighting both effectiveness and robustness. This work advances the development of reliable MLLMs for real-world document analysis by addressing critical challenges in visual-linguistic alignment under degradation. Data is available at `https://huggingface.co/datasets/bytedance-research/KIE-HVQA`.

## 1   Introduction

In recent years, there have been significant advancements in MLLMs [1, 9, 33, 4] for document understanding [11, 18]. These models integrate textual semantics with visual features, offering new paradigms for automated processing of identity cards, invoices, contracts, and similar applications.

MLLMs demonstrate near-human performance in documents understanding across several domains. Enhancements in language models have improved multilingual support and incorporated prior knowledge, leading to more accurate text parsing. Advancements in visual encoders [31, 16, 15], such as increased resolution, have enhanced the ability to capture image details. Additionally,

39th Conference on Neural Information Processing Systems (NeurIPS 2025).

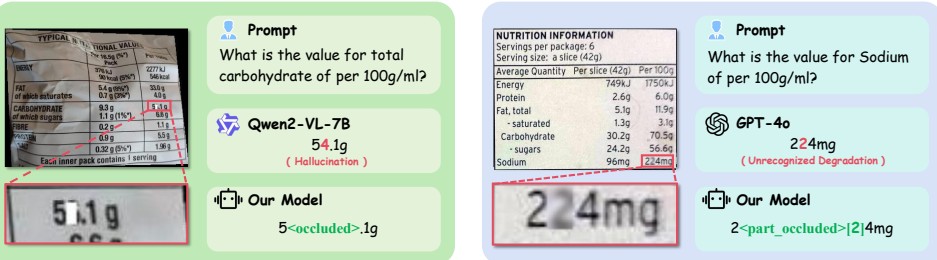

Figure 1: The performance of Qwen2.5-VL-7B (*left*) and GPT-4o (*right*) in interpreting degraded text images. The Qwen2.5-VL-7B model may experience hallucinations, identifying values not present in the image, while the GPT-4o model struggles with unrecognized degradation due to partial text occlusion. Previous models have not thoroughly addressed these issues, whereas the model proposed in this paper effectively resolves them, providing more accurate recognition results.

refined layout-based instructions [35, 20] have enabled systems to better understand document structures. In particular, current MLLMs [4, 40] exhibit strong cross-modal reasoning capabilities when working with high-quality images and standardized layouts. However, current research has not yet fully addressed the incompleteness within their paradigm in real-world scenarios. The core challenge stems from models' inability to enforce strict adherence to visual signals. When confronted with practical complexities—including image blurring or unconventional formatting—the models frequently generate cross-modal hallucinatory content that deviates substantially from input data.

The *OCR hallucination issue* in MLLMs stems from three critical challenges across the model development process. *First*, during the pre-training phase, there is a significant lack of key information extraction (KIE) data and clear annotations related to degraded visual scenarios[32], which limits the model's ability to process challenging visual inputs. *Second*, in the instruction fine-tuning phase, the paradigm for handling degraded visual scenarios is often overlooked, as researchers generally assume OCR tasks involve non-degraded inputs [17, 8, 21, 14]. Even MLLMs with strong visual capabilities fail to demonstrate the necessary reasoning abilities for real-world degraded documents. *Third*, in the evaluation phase, the absence of dedicated benchmarks for quantifying OCR hallucination in document understanding tasks impedes progress, as the field lacks both comprehensive metrics and sufficient annotated data due to the inherent challenges in collecting and labeling degraded samples. As a result, when confronted with visually compromised inputs like glare-obstructed identity cards or low-contrast reports, models exhibit cognitive bias by defaulting to linguistic priors rather than anchoring decisions to observable visual evidence, leading to potentially catastrophic misinterpretations in critical applications [6]. The examples are illustrated in Figure 1.

To address these pressing challenges, this paper introduces a comprehensive benchmark and a novel framework designed to tackle the critical issues of vision-faithful reasoning in degraded document understanding. We present KIE-HVQA, the first benchmark specifically designed to evaluate OCR hallucination under real-world noise conditions. This dataset includes 2,000 annotated training samples and 400 rigorously curated test instances spanning diverse document types, including identity cards, receipts, and invoices. Each sample is carefully designed to simulate real degradation scenarios, such as motion blur and low contrast, necessitating fine-grained visual-textual alignment for accurate key information retrieval. For instance, the task may involve extracting ID numbers from partially occluded cards or resolving ambiguous dosage entries in faded prescriptions.

Inspired by the successful practices of reinforcement learning in computer vision tasks [40, 19], we employ reinforcement learning as a tool to provide a feasible approach to addressing this issue. Unlike typical MLLM tasks such as VQA [2], the KIE [39] task benefits from having quantifiable standard answers, which allows for the construction of precise foundational rewards and the design of appropriate rewards for various degradation scenarios. By employing Group Relative Policy Optimization (GRPO) algorithm [10], we can supervise the model to enhance its existing OCR capabilities and develop a self-reflective KIE instruction paradigm that addresses visual degradation. This approach encourages models to prioritize visual evidence over linguistic priors, ensuring that decisions are more robustly anchored to observable data, marking a significant advancement in overcoming the challenges of vision-faithful reasoning and cross-modal OCR hallucination.

To validate the efficacy of our training methodology and dataset, we implemented our proposed approach to enhance Qwen2.5-VL [4] and conducted comprehensive experiments to benchmark our method against state-of-the-art multimodal models. Our method achieves a notable ∼28% absolute improvement in hallucination-free accuracy on the KIE-HVQA benchmark. The contributions of this paper are summarized as follows:

- We propose KIE-HVQA, the first benchmark for evaluating hallucinations in degraded documents. This benchmark simulates real-world degradations with pixel-level annotations and OCR reliability scores, enabling a comprehensive assessment of OCR hallucinations under degraded conditions.

- Based on the characteristics of the KIE task, we designed precise reward modeling for the GRPO algorithm. By integrating this with an appropriate coldstart, we successfully enhanced the model's ability to reason effectively with degraded visual input, significantly reducing hallucination without sacrificing its original OCR capabilities.

- Through extensive experiments, our model demonstrates superior reasoning capabilities. Our model achieves a ∼28% improvement in hallucination suppression compared to GPT-4o.

## 2 Related Work

### 2.1 Reasoning in Multimodal Large Language Models

Recent developments in large language models (LLMs) [1, 30, 5] demonstrate that simulating human-like thought processes and implementing sequential reasoning strategies can significantly improve performance on complex problem-solving tasks. A significant innovation [10, 29] involves DeepSeek-R1's implementation of extensive reinforcement learning [22, 24] techniques to foster self-evolving cognitive pathways in LLMs, substantially enhancing their performance on sophisticated reasoning challenges. Inspired by advancements in LLM reasoning, researchers [36, 7, 40, 28] have applied CoT prompting and developed SFT datasets with step-level reasoning for MLLMs.

Despite recent advancements, there remains a paucity of research focused on applying reasoning to OCR tasks, particularly in addressing hallucination issues. To our best knowledge, our approach is the first to utilize RL training to effectively tackle hallucination problems in OCR tasks.

### 2.2 OCR benchmarks

In the early era of deep learning, a variety of specialized benchmarks emerged to address different challenges, such as natural-scene text [13], web-scene text [25], and multi-directional and curved text recognition. The current OCRBench [17, 8, 37] for evaluating MLLMs primarily targets line-granularity recognition. Other benchmarks, such as DocLocal4K [11] and FOX [14], curate data mainly from document images.

Currently, these OCR benchmarks predominantly focus on document understanding and key information retrieval, often neglecting issues such as hallucinations and misrecognitions caused by image degradation. Our newly proposed benchmark KIE-HVQA takes a significant step forward by addressing these overlooked challenges for the first time.

## 3 KIE-HVQA

### 3.1 Task Description

To provide a comprehensive evaluation framework for degraded document hallucination tasks, the KIE-HVQA benchmark introduces a visually-grounded question answering task. This task demands precise alignment between textual semantics and degraded visual evidence in real-world documents. When presented with a degraded document image, such as a blurred ID card or occluded images, and a question, models are required to perform several key tasks. Initially, they must identify text elements relevant to the question through multi-modal grounding. Subsequently, they need to assess recognition confidence at the character level by analyzing edge sharpness, measuring contrast ratios, and verifying contextual coherence. Finally, models should generate answers based on visually verifiable content, while clearly indicating regions of uncertainty.

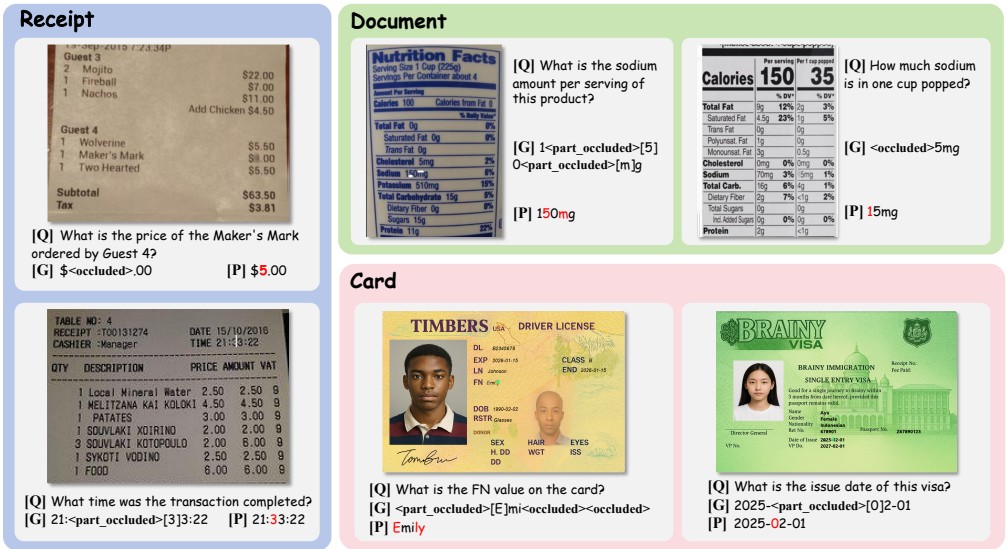

Figure 2: Visualization of the three types of data in our KIE-HVQA benchmark. [Q] represents the question, [G] denotes the ground truth, and [P] indicates the prediction generated by Qwen2.5-VL with zero-shot prompt. The data exhibit varying degrees of degradation, such as blurriness or damage, which affect the model's predictive accuracy.

The benchmark focuses on evaluating models' ability to minimize reliance on parametric knowledge biases in situations of partial legibility, such as medical prescriptions where dosage units are clear, but frequencies are not. This requires that models adhere strictly to the available visual evidence, ensuring accurate interpretation and response based on the information present in the document.

## 3.2 Annotation Curation

This section presents the curation of annotations in three stages: dataset collection, instruction formulation, and manual verification of results.

To tackle the challenge of vision-faithful reasoning in degraded document understanding, we assembled a diverse dataset from three main sources: OCRBench [8], WildReceipt [27], and GPT-4o-generated images. Each source was carefully chosen to simulate realistic degradation scenarios and test the robustness of MLLMs.

**OCRBench.** Our methodology is benchmarked on 100 key information queries from OCRBench. We first employ a text detection model combined with a character localization model [3], trained under a weak supervision framework, to extract the precise coordinates of characters in the answers. Given the potential randomness in the reading order of image characters, we utilize GPT-4o to reconstruct the correct character sequence. Subsequently, these coordinates undergo a random degradation process.

To ensure the accuracy of the degradation results and prevent model hallucinations, we established a rigorous dual-model evaluation mechanism. This process involves cropping each character from its degraded region in the original image and inputting these sub-images into two leading Multimodal Large Language Models (MLLMs)—GPT-4o and Qwen2.5-VL-72B—for parallel verification. This model pairing was strategically chosen to leverage their complementary strengths: GPT-4o serves as the performance benchmark for closed-source models, excelling in complex visual pattern recognition, while Qwen2.5-VL-72B is a state-of-the-art open-source model for OCR tasks. Each MLLM independently provides a binary judgment (visible/invisible), and any discrepancies between their outputs are resolved through expert human review. This design mitigates bias from any single model and ensures the accurate annotation of borderline cases, providing a solid foundation for subsequent character-level reliability annotations. The detailed format of our answer data is presented in Section 4.

**WildReceipt.** From the WildReceipt dataset, we extracted entity-type answers from the original dataset and used MLLMs to generate corresponding questions. The images were modified using the same techniques applied to OCRBench, and the answers were reconstructed in a similar manner.

**GPT-4o-generated Images.** We used GPT-4o image generator [1] to create 200 synthetic templates of IDs and documents with fictional information, ensuring privacy compliance. The information on these IDs was generated by GPT-4o and then added using Photoshop. Based on this, we designed corresponding question-and-answer pairs. To evaluate the model's performance in handling complex visual information, we applied degradation techniques to the answers, including adding obfuscation and blur effects.

We provide some samples from the KIE-HVQA dataset in Figure 2. This comprehensive dataset allows for rigorous testing of the ability to maintain visual-textual alignment and avoid hallucinations, even under conditions of visual degradation.

### 3.3 Evaluation Criteria

To assess the performance of our model in understanding degraded documents, we have developed three comprehensive evaluation metrics. These metrics are designed to capture various aspects of OCR performance under different visual conditions.

**Legible Character Accuracy.** This metric measures the character-level accuracy in regions of the document with high visibility. It serves as a indicator for the model's basic OCR capabilities, reflecting its ability to accurately recognize text in ideal conditions. A high score indicates that the model can perform near-perfect recognition when the text is clearly visible, thus setting a standard for its performance under optimal conditions.

**Degraded Character Accuracy.** This metric evaluates the recognition accuracy in predefined regions that have been annotated as degraded due to factors such as motion blur, occlusion, or low contrast. It is specifically designed to test the model's robustness against visual ambiguities and its ability to maintain accuracy in challenging conditions. For words that are degraded but do not pose a risk of hallucination, the model should output the corresponding characters. However, for areas with a high risk of hallucination, the model should demonstrate awareness to appropriately reject providing an answer. This metric ensures that the model can effectively handle and interpret text in degraded circumstances while minimizing errors due to hallucinations.

**Global OCR Performance.** Focuses on task-specific text extraction quality through two critical metrics: Accuracy of OCR results for question-critical text regions referenced in VQA answers and Normalized Levenshtein distance between the OCR-extracted text and ground truth specifically for information required to answer the question.

## 4 Method

In this section, we systematically model the OCR hallucination issue as a fundamental problem with precise rewards, reflecting various degradation issues through different reward functions. We then extend this framework by introducing a new reward paradigm and aligning model behavior using reinforcement learning. In Section 4.1, we provide an overview of the rule-based GRPO algorithm, which serves as the basis for our approach. In Section 4.2, the cold-start initialization method and the data generation process are elaborated in detail. In Section 4.3, we present the GRPO algorithm and the degradation-based OCR reward function, explaining their roles in the training process.

### 4.1 Preliminaries

The training of DeepSeek-R1 [10] employs Group Relative Policy Optimization (GRPO), a novel reinforcement learning algorithm that differs from conventional methods like PPO [23]. GRPO assesses strategies by comparing groups of generated responses, eliminating the need for a critic model and simplifying the training process.

For a given input $q$, GRPO generates $G$ responses $\{o_1, o_2, \ldots, o_G\}$ using the current policy $\pi_{\theta_{old}}$. It then evaluates each response via a predefined reward function to obtain rewards $\{r_1, r_2, \ldots, r_G\}$. To

---

[1]https://openai.com/index/introducing-4o-image-generation/

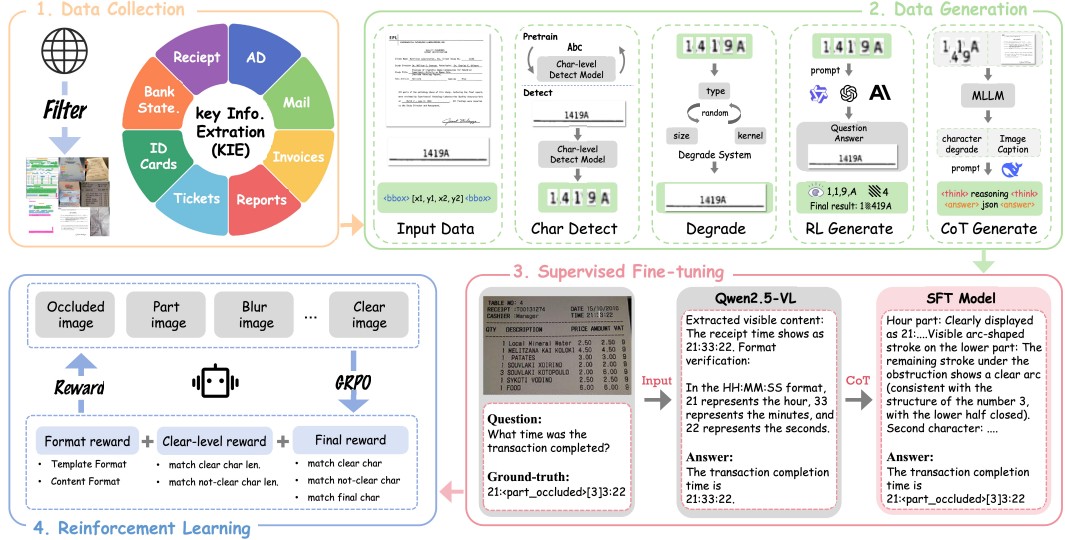

Figure 3: Overview of our framework: The cross-modal reasoning pipeline begins with comprehensive data collection, incorporating visual formal descriptions to enhance reasoning capabilities. We utilize a multimodal approach to generate training data. After training, the process includes supervised fine-tuning to address OCR hallucination issues and improve reasoning accuracy. Finally, rule-based reinforcement learning is applied to enhance generalization across multimodal tasks.

determine the relative quality of each response, GRPO normalizes the rewards:

$$A_i = \frac{r_i - \text{mean}(\{r_1, \ldots, r_G\})}{\text{std}(\{r_1, \ldots, r_G\})}, \tag{1}$$

In the training procedure, GRPO initializes a trainable policy model $\pi_\theta$ and a frozen reference model $\pi_{ref}$. The policy model $\pi_\theta$ is optimized by maximizing the following objective function of $G$.

$$\mathcal{J}\text{GRPO}(\theta) = \frac{1}{N}\sum_{i=1}^{N}\left(\frac{\pi_\theta(o_i|q)}{\pi_{\theta_{\text{old}}}(o_i|q)}A_i - \beta \cdot \mathcal{KL}(\pi_\theta(o_i|q)\|\pi_{\text{ref}}(o_i|q))\right) \tag{2}$$

Here, $N$ represents the number of completions in a group, and $\beta$ is a hyperparameter. This objective function encourages the model to prioritize completions with higher advantages within the group while maintaining proximity to the initial model.

## 4.2 Cold-start Initialization

Recent studies have focused on developing multimodal reasoning datasets that build upon existing fine-tuned data, with the objective of enhancing the reasoning capabilities of MLLMs and improving their overall performance. This paper built a multimodal CoT-OCR dataset that encompasses complex OCR degradation scenarios, enabling models to reason in a human-like manner. Several reasoning models, such as DeepSeek-R1 [10] and Kimi K1.5 [29], already possess the capability to perform natural cognitive processes using CoT reasoning. These models can generate high-quality CoT data that includes human-like self-reflection processes. However, these models are purely language-based and cannot directly process multimodal data to produce CoT data.

To address the challenge of processing multimodal data with language-based models, we integrate existing MLLMs with DeepSeek-R1. First, we convert multimodal information, such as images and text, into purely textual information using GPT-4o. This involves inputting image-question-answer pairs and prompts into GPT-4o to generate a pseudo-CoT that includes both image descriptions and reasoning processes. Next, we merge these image-question pairs with the generated pseudo-CoT and prompts, and feed them back into the MLLM to produce detailed image descriptions. These descriptions are then combined with the textual information and input into DeepSeek-R1, allowing it

to execute a high-quality CoT process. This approach ensures that the resulting CoT data captures complex reasoning in a way that mimics human cognitive processes.

Finally, we pair the pure textual CoT data generated by DeepSeek-R1 with the corresponding images to create an integrated multimodal CoT dataset for cold-start initialization, as illustrated in the data generation process shown in Figure 3. The CoT data obtained through this method closely aligns with human cognitive behavior, allowing the reasoning process to exhibit natural and logical thinking.

## 4.3  RL with OCR reward

We implement GRPO with hard formatting result rewards to enhance the self-learning capabilities of the model. For each question $q$, GRPO samples a group of generated output set $\{o_1, o_2, \cdots, o_G\}$ from policy model $\pi_{\theta_{old}}$. Then GRPO maximizes the objective function in Eqn. 2 and optimizes the model $\pi_\theta$. Specifically, we introduce a rule-based reward for degraded OCR scenarios. This reward function is designed to ensure that OCR models maintain fidelity to visual input when generating textual output. It is specifically tailored to handle varying levels of character clarity within visual data, categorizing them into three distinct cases for accurate recognition and transcription. The criteria and objectives of our reward function are as follows:

*Legible Character*: For characters that are entirely clear and unambiguous, the model is required to accurately recognize and retain these characters in the final OCR output. This ensures that any fully legible text is preserved without alteration.

*Partially Obscured but Human-Recognizable Characters*: In cases where the characters are partially obscured or blurred but still recognizable by a human observer, the model should identify these as "anomalous" characters. Although these characters lack perfect clarity, they must be included in the final OCR output, reflecting the human ability to infer their identity.

*Unrecognizable Characters*: Characters that are entirely obscured and cannot be identified should not be included in the OCR output. Instead, these should be represented by a space to prevent any hallucination or erroneous inference by the model.

For the degraded input text in Figure 4: "B<part_occluded>[e]au<occluded>[ti]ful".
The reward function enforces visual-textual fidelity through a multi-stage analytical process. During the character-level classification phase, clear characters ("*B, a, u, f, u, l*") are preserved verbatim, moderately blurred characters such as "*e*" (marked with tags) are retained as partially occluded anomalies, while severely obscured character clusters like "*t, i*" are classified as unrecognizable units.  Building upon these assessments, quantitative evaluations yield 6 legible characters, 1 partial occlusion instance, and 2 completely obscured characters.

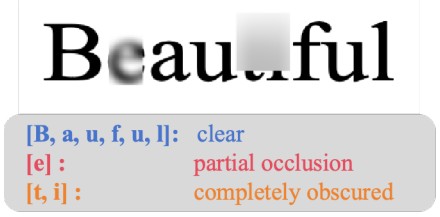

Figure 4: The figure illustrates the degrade criteria of each letter in the word "Beautiful". The letters "*B, a, u, f, u, l*" are clearly visible; the letter "*e*" is partially occluded; the letter "*t, i*" is completely obscured.

This reward function is integrated into the GRPO training objective to systematically guide the learning process of the model. The reward calculation is formalized as:

$$F = \sum_i f_i(A_i, G_i) \cdot \left(1 - \sum_j g_j(A_j, G_j)\right) \tag{3}$$

Here, $i$ represents different categories or types of OCR results, including clear characters, unclear characters, and the final answer. $j$ involves counting specific elements within the OCR results. $A$ and $G$ represent the model's output and the ground truth, respectively. $f$ and $g$ denote the OCR edit distance evaluation metric and the numerical calculation evaluation metric, respectively. The structured reward signals enable precise alignment between visual faithfulness and textual accuracy, with error type differentiation driving targeted model improvement. The training loop continuously evaluates the model's performance against a diverse set of degraded OCR samples, allowing for iterative improvement.  As the model encounters varied visual challenges, the reward function dynamically adjusts, promoting adaptability and robustness in handling real-world OCR scenarios.

**Algorithm 1** Reward Function for OCR Task

```
 1: function CALCULATE_METRICS(pred, truth)
 2:     if len(pred) = 0 and len(truth) = 0 then
 3:         return 1.0
 4:     end if
 5:     edit_dist ← levenshtein_distance(pred, truth)
 6:     max_len ← max(len(pred), len(truth))
 7:     similarity ← 1 − (edit_dist/max_len)
 8:     return similarity
 9: end function
10: function REWARD(answer, gt)
11:     (not_clear_metric, final_metric, clear_metric) ← calculate_metrics(answer, gt)
12:     bs ← c₁ × not_clear_metric + c₂ × clear_metric + c₃ × final_metric
13:     return bs
14: end function
```

| Models | OCRbench-KIE subset | | | | Wildreceipt subset | | | | Card subset | | | | Average | | | |
|---|---|---|---|---|---|---|---|---|---|---|---|---|---|---|---|---|
| | Clr | Nc | Final | Avg | Clr | Nc | Final | Avg | Clr | Nc | Final | Avg | Clr | Nc | Final | Avg |
| GPT-4o (1120) [1] | 24.41 | 34.18 | 29.39 | 29.33 | 18.17 | 34.61 | 28.55 | 27.11 | 33.86 | 41.86 | 42.28 | 39.33 | 22.78 | 36.13 | 31.74 | 30.21 |
| Claude3.5-Sonnet | 24.30 | 29.49 | 27.13 | 26.97 | 23.13 | 18.75 | 24.54 | 22.14 | 30.24 | 22.92 | 29.98 | 27.71 | 24.92 | 21.63 | 26.22 | 24.25 |
| Claude3.7-Sonnet | 25.76 | 40.63 | 33.95 | 33.45 | 15.52 | 31.2 | 23.57 | 23.43 | 26.32 | 34.87 | 26.77 | 29.32 | 19.77 | 33.73 | 26.17 | 26.56 |
| Gemini2.5-pro | 55.34 | 53.85 | 56.18 | 55.12 | 27.33 | 24.22 | 29.13 | 26.89 | 47.71 | 46.94 | 26.77 | 40.47 | 36.94 | 34.64 | 33.53 | 35.03 |
| InternVL3-8B [43] | 4.05 | 4.42 | 4.09 | 4.19 | 7.26 | 7.61 | 7.46 | 7.44 | 12.49 | 16.65 | 11.22 | 13.45 | 7.83 | 9.03 | 7.68 | 8.18 |
| InternVL3-38B [43] | 9.34 | 14.29 | 9.00 | 10.88 | 10.97 | 18.93 | 11.20 | 13.70 | 21.96 | 26.45 | 21.03 | 23.15 | 13.11 | 19.75 | 12.98 | 15.28 |
| InternVL3-78B [43] | 1.90 | 2.00 | 2.00 | 1.97 | 5.00 | 7.60 | 6.00 | 6.20 | 12.49 | 16.65 | 11.22 | 13.45 | 6.09 | 8.59 | 6.43 | 7.04 |
| Qwen2.5-VL-8B [4] | 29.08 | 37.68 | 26.84 | 31.20 | 14.69 | 19.16 | 15.66 | 16.50 | 26.95 | 26.70 | 27.74 | 27.13 | 20.02 | 24.19 | 20.37 | 21.53 |
| Qwen2.5-VL-32B [4] | 28.54 | 42.31 | 29.43 | 33.43 | 10.34 | 30.12 | 10.66 | 17.04 | 23.86 | 32.28 | 23.65 | 26.60 | 16.64 | 32.81 | 16.95 | 22.14 |
| **Our Model + SFT** | 52.41 | 68.33 | 51.02 | 57.25 | 50.52 | 57.01 | 49.21 | 52.25 | 45.03 | 48.78 | 50.01 | 47.94 | 49.65 | 57.25 | 49.72 | 52.20 |
| **Our Model + SFT+RL** | **57.52** | **74.03** | **57.59** | **63.05** | **56.54** | **59.31** | **58.41** | **58.09** | **50.82** | **56.38** | **54.29** | **53.83** | **55.45** | **61.34** | **57.35** | **58.05** |

Table 1: Evaluation results of closed-sourced, open-sourced and our models on KIE-HVQA benchmark. Clr, Nc, Final, Avg represent clear characters, not clear characters, final OCR and average results, respectively.

In summary, this enhanced GRPO framework, featuring a novel reward function, enables a more balanced training approach. By guiding the model to prioritize both high textual accuracy and faithfulness to visual input, this methodology leads to more reliable and trustworthy outputs.

## 5 Experiment

### 5.1 Experiment Settings

**Training Dataset.** To obtain the cold-start dataset, we created custom data by generating word images using random fonts and varying degrees of degradation. We utilized the bounding boxes from TextOCR [26] to acquire relatively accurate character-by-character coordinates, thereby generating a set of cold start data with a "think" phase. In the GRPO phase, we mixed part of TextOCR [26], WildReceipt [27], and other OCR datasets [12, 34] as our reinforcement learning training dataset.

**Implementation Details.** For the cold-start dataset preparation, we utilized GPT-4o and the reasoning LLM DeepSeek-R1. We then processed the VQA datasets using GPT-4o and DeepSeek-R1 over approximately 12 hours. For the cold-start initialization, we used Qwen-2.5-VL-7B-Instruct as the base model and performed supervised fine-tuning for 5 epochs with a learning rate of 1e-6 and a data rollout batch size of 512. This process required approximately 4 hours, using the LLaMA-Factory framework [42]. Following the cold-start phase, we trained the model using the collected dataset with the GPRO method over several hours, employing the Easy-R1 framework [41].

### 5.2 Main Results

Table 1 provides a detailed evaluation of document understanding performance on the KIE-HVQA benchmark. Our model sets a new standard with an average distance score of 58.05%, outperforming close-sourced models GPT-4o, Claude and Gemini. This substantial improvement underscores our

model's superior ability to maintain visual-textual alignment even under challenging degradation conditions.

In scenarios simulating partial occlusion, our model achieves a remarkable 61.34% accuracy in not clear character-level OCR distance evaluations, surpassing GPT-4o's 36.13%. This success is attributed to our uncertainty-aware grounding mechanism, which effectively reduces hallucination.

The results validate the robustness of our approach across various metrics, including average distance, clear character recognition, and handling of occluded text. Our model demonstrates balanced performance across all dimensions, proving its capability to adapt to different levels of text degradation and complexity. Crucially, these findings highlight a path towards more robust and trustworthy OCR by addressing the inherent limitations of previous methods. Rather than simply relying on feature-based estimation prone to critical errors under degradation, this work enables a more nuanced understanding and processing of real-world documents, particularly in challenging scenarios

### 5.3 Abalation Study

**Analysis of Training Strategy.** To evaluate the effectiveness of training data, we compared the model's performance under two training strategies: (1) applying Supervised Fine-Tuning on our dataset, and (2) optimizing the SFT-trained model with Reinforcement Learning. As shown in Table 1, SFT significantly improved the model's performance on the KIE-HVQA benchmark while applying RL afterward led to additional performance gains, enabling the model to tackle more complex problems. This study demonstrates that our training data is crucial for enhancing model performance, and the combination of SFT and RL is a powerful and effective strategy for maximizing reasoning and thinking capabilities in KIE-HVQA

**Analysis of General OCR Capability Preservation.** To investigate whether our enhanced degradation handling will affect general OCR capabilities, we conducted comparative evaluations in three standard OCR domains of OCRbench [17]: Scene Text-centric VQA, Doc-oriented VQA, and Key Information Extraction. As shown in Table 2, our model achieves comparable performance to specialized baseline models. This demonstrates that our uncertainty-aware grounding mechanism specifically targets degraded regions without affecting general text recognition capabilities.

| Model | Scene | Doc | Info |
|---|---|---|---|
| GPT-4o (1120) [1] | 180 | 167 | 163 |
| Claude3.7-Sonnet | 159 | 130 | 125 |
| Qwen2.5-VL-7B [4] | 181 | 181 | 182 |
| MiniCPM-o-2.6 [38] | 187 | 182 | 187 |
| Our Model | 180 | 179 | 183 |

Table 2: Ablation studies demonstrating the preservation of general OCR capabilities.

**Analysis of Reward Setting.** Ablation studies on our dataset demonstrate the necessity of integrating composite rewards, as shown in Table 3. The format reward primarily ensures that the model adheres to the expected format in its responses. Therefore, the ablation experiments focus mainly on the character matching aspect of the final reward. When considering only the clear character reward, the model's

| Reward Setting | Clr | Nc | Final |
|---|---|---|---|
| only clear | 50.64 | 44.15 | 53.34 |
| only final | 51.06 | 54.06 | 54.24 |
| all rewards | **55.45** | **61.34** | **57.35** |

Table 3: Ablation on reward setting.

performance on not clear characters significantly declines. Similarly, when focusing solely on the final character reward, the results are inferior compared to the combination of all rewards. Our framework significantly outperforms single-reward variants, showing marked improvements across all evaluation dimensions. This validates that multi-objective reward synthesis is crucial for handling real-world document degradation patterns.

## 6 Conclusion

This paper addresses the challenge of cross-modal OCR hallucination in degraded document understanding by introducing KIE-HVQA, the first benchmark designed to evaluate vision-faithful reasoning under real-world noise conditions. Our benchmark simulates practical degradation scenarios, facilitating a comprehensive assessment of MLLMs' performance in challenging environments. We propose a novel GRPO-based framework with a multi-objective reward mechanism to enforce

vision-faithful reasoning. This framework incorporates uncertainty-driven rejection behaviors, effectively suppressing hallucinations in ambiguous regions and enhancing adaptability to complex tasks. Extensive experiments demonstrate the efficacy of our approach, with our 7B-parameter model achieving $\sim$28% absolute improvement in hallucination-free accuracy over GPT-4o on the KIE-HVQA benchmark. This highlights our model's robustness and computational efficiency in maintaining visual-textual alignment under visual degradation.

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
