# OpenReview forum: "Seeing is Believing? Mitigating OCR Hallucinations in Multimodal Large Language Models"
_NeurIPS.cc/2025/Conference — NeurIPS 2025 poster_

### Official Review · Reviewer_A3fL · 2025-06-21

**Clarity:** 2
**Significance:** 3
**Originality:** 4
**Rating:** 4
**Confidence:** 5

**Summary:**

This paper aims to address MLLM’s OCR hallucination issue in degraded document understanding. The authors propose a dataset named KIE-HVQA, consisting of various documents with real-world degradation and pixel-level annotation. For annotation, they tag the degraded area (blur, occlusion, low contrast) using special tokens like `<part_occluded>` and `<occluded>`, following the correct content. They also propose a new RL-based KIE MLLM, which is trained using GRPO and self-designed rewards. The authors conduct experiments to validate their methods’ effectiveness.

**Questions:**

For rebuttal questions, please see weaknesses. The major concern of this paper lies in the lack of many critical model implementation and experimental details. If the authors can address these issues, I will consider raising my score.

**Ethical Concerns:**

["NO or VERY MINOR ethics concerns only"]

**Final Justification:**

Most of the concerns are addressed. Therefore, I raise the rating to borderline accept.

**Limitations:**

Yes.

**Paper Formatting Concerns:**

No.

**Quality:**

2

**Strengths And Weaknesses:**

> Strengths:

1. The authors discover a novel perspective: the hallucination of OCR on degraded documents. This is interesting and can encourage the community to pour more attention into this underexplored issue.
2. The KIE-HVQA dataset is a valuable contribution to the community. There is a lack of document analysis datasets with such real-world, character-level degradation. This pioneering effort fills this gap and potentially contributes to broader applications in the document field beyond this paper’s task.
3. The KIE-HVQA includes diverse data sources, including receipts, documents, and ID cards. The authors generate ID card images using GPT-4o to avoid privacy issues. This is an effective and important approach to producing privacy-sensitive data.
4. In Sec. 4.2, the authors generate CoT data using an iterative approach, which is reasonable for producing high-quality CoT data.

> Weaknesses:

1. In Sec. 3.2, L141-142, the authors only say they use GPT and Qwen2.5-VL-72B to evaluate the degraded results, without giving a concrete evaluation process. Why are two models used here? How do they operate? The authors should provide more details.
2. In Sec. 3.3, how do the authors implement the measurement of legible and degraded character accuracy? The authors provide only definitions without implementation details.
3. In Fig. 3, the bottom-right panel “3. Supervised Fine-Tuning” is unclear and confusing. What is the role of Qwen2.5-VL here? SFT is a training procedure. If it takes the QA pair as input, it looks more like a data generation procedure, which should not occur during training. In addition, what does the “CoT” output by Qwen2.5-VL mean? I think the SFT model is trained using the CoT data generated during “2. Data Generation”, and the CoT here makes me confused. Can the authors provide some clarifications and make Fig. 3 clearer?
4. The title of Sec. 4.2 is “Cold-start Initialization”. As far as I know, cold-start initialization refers to supervised fine-tuning (SFT) on certain data. Yet, this sub-section mainly describes the CoT data generation process rather than the training process. If the authors aim to describe data generation, then they should change the title. Also, it seems that the authors have omitted the illustration of SFT (despite its presence in Fig. 3) since Sec. 4.3 is about RL training. An SFT-related sub-section should precede RL training.
5. In Sec. 4.3, the authors define three rewards (legible character, partially obscured but human-recognizable characters, unrecognizable characters), but do not provide their detailed implementations. Algorithm 1 appears to compute rewards; however, the definitions inside do align with the three rewards (the outputs’ names are not_clear_metric, final_metric, and clear_metric, which are different from the above three). The authors should give necessary details for reward computing.
6. For Algorithm 1, the CALCULATE_METRICS function simply outputs one similarity. However, in line 11 of this algorithm, this function outputs three values. This inconsistency is confusing.
7. In experiments, what prompts are used in testing the closed-source models (e.g. GPT, Claude)? Are the prompts the same as those used for training the proposed model to ensure fairness?
8. In Table 1, the proposed model exhibits better performance than other models not trained on the proposed dataset. However, for recognition of those occluded texts, what is the rationale behind the outperformance of the proposed models? Recognizing visual text mainly depends on the ability of the visual encoder. Since the authors did not specifically improve visual modeling and text is occluded, performance on these texts should be close. Can the authors provide a deeper analysis of such outperformance?
9. In L322-323, what does the format reward refer to? I did not found any definition of the format reward in either the methodology section or the experiment section.
10. typo: L106: MLMMs -> MLLMs; L145: We -> we; L307: KIE-HVQA -> KIE-HVQA. L320: setting -> Setting

---

> ### Author Rebuttal · Authors · 2025-07-31
>
> **Q1，Q2:**
>
>   Thank you for your comment. We will clearly specify these details in the final version. In the degradation results verification phase (L141-L144), we established a rigorous dual-model evaluation mechanism. This process begins with a character localization model extracting the coordinates of degraded regions, followed by cropping each degraded character into a sub-image and inputting it into GPT-4o and Qwen2.5-VL-72B for parallel verification (Lines 141-142). These two models were selected based on their complementary strengths: GPT-4o serves as the performance benchmark for closed-source multimodal models and excels in complex visual pattern recognition, while Qwen2.5-VL-72B is currently the leading open-source model in OCR capability. Each model independently outputs a binary judgment (visible/invisible), and discrepancies between their judgments are resolved through expert human review. This design avoids bias from any single model and ensures accurate annotation of borderline cases through human intervention, providing a solid foundation for subsequent character-level reliability annotations.
>
>  Regarding the implementation of legibility measurement (Section 3.3), we adopt a multi-model consensus mechanism as the core criterion. Specifically, when both GPT-4o and Qwen2.5-VL-72B correctly recognize a character independently (i.e., without relying on contextual inference), the character is labeled as "Partially Obscured but Human-Recognizable Characters"; when both models fail, it is marked as an "Unrecognizable Characters." All model outputs are ultimately verified and refined through expert human review to ensure annotation accuracy.
>
> **Q3，Q4:**
>
> Yes, there is some ambiguity in the illustration of Figure 3. Specifically, Qwen2.5-VL and SFT MODEL refer to the baseline reference model and the cold-started model, respectively. The contents shown correspond to the original response generated by these models and the chain-of-thought (CoT) outputs from the cold-started model. It should be clarified that CoT here does not refer to the model outputs themselves, but rather to the training data generated in the data generation phase used for supervised fine-tuning (SFT). This design intuitively demonstrates the training effect through comparison, avoiding confusion with the data generation process in Stage 2.
>
> Regarding Q4, we appreciate your precise correction of the term "cold-start initialization." We will clearly distinguish between the data generation and training phases in the main text and revise the figure accordingly: first, a dedicated dataset is constructed via the multimodal CoT data generation pipeline (currently described in Section 4.2), followed by an independent subsection detailing the SFT process. Specifically, we performed supervised fine-tuning (SFT) on the pretrained Qwen2.5-vl-7B model as the foundational multimodal large language model (MLLM) for cold-start initialization. The model after cold-start initialization is named the SFT MODEL. At this stage, the base MLLM has already learned complex reasoning patterns from DeepSeek-R1; however, this also leads to an "overthinking optimization problem," where the SFT MODEL tends to generate unnecessarily lengthy reasoning processes on certain problems, whereas correct reasoning typically occurs within shorter cognitive chains. Based on this, the model requires additional GRPO training to further refine its performance.
>
> **Q5:**
>
> Thank you for your comment. We have designed a corresponding reward mechanism for the three different categories of characters. For clearly recognizable characters, such as obvious text, we use the clear_metric for evaluation, which requires the model’s output to exactly match the visual input. For partially occluded characters that both MLLM and humans can still infer, we apply the not_clear_metric, which allows the model to retain labeled anomalous characters, reflecting a certain degree of tolerance.
>
> As for completely unrecognizable characters, the model’s performance is indirectly constrained by the **final_metric**. For example, in the word “Beautiful” shown in Figure 4, the letters **t** and **i** are severely blurred and cannot be recognized. If the model attempts to guess and output specific content for these characters, the overall OCR similarity score will significantly decrease, thereby penalizing the model and encouraging it to avoid generating fabricated or uncertain information in such regions.
>
> **Q6:**
>
> Sorry for the confusion caused. The contradiction you observed in the function output actually arises because both pred and truth inputs are triplets, and consequently, the function returns a triplet as well. This is a Pythonic style of handling multiple related values together. To avoid any ambiguity, we will provide a clear and detailed definition of this triplet input data structure in the revised version.
>
> **Q7:**
>
> The prompts used are detailed in the appendix. Unlike the prompts used during model training, we specifically added zero-shot output format examples to prevent the model from producing incorrect responses due to a lack of clear format guidance. In fact, without providing these examples, the model’s responses often exhibit format inconsistencies or fail to meet the expected requirements. For instance, in the following case, the ground truth is: {"clear Char-level OCR": "2 . 0", "not clear enough Char-level OCR": "", "clear number": 3, "not clear enough number": 0, "Final OCR": "2. 0"}, while the model’s response is: "The price of the "NIVEA FOR MEN" on the receipt is £0.94." This demonstrates that without format examples, the model tends to deviate from the expected output format.
>
> **Q8:**
>
> Thank you for your insightful discussion on this important issue. Your observation is highly accurate — the performance advantage of our model in recognizing occluded text does not stem from improvements in the visual encoder, but rather from our innovative approach to addressing OCR hallucination problems. The key reason why traditional models perform poorly in degraded scenarios lies in their excessive reliance on LLMs, which leads to hallucinations. Our method fundamentally overcomes this challenge through three key innovations:
>
> First, the character-level reliability assessment mechanism enables the model to finely distinguish readable information from noise interference. As shown in Figure 4, when processing the word “Beautiful,” the model analyzes each character independently: clearly visible letters “B, a, u, f, u, l” are directly retained; the partially occluded “e” is marked with low confidence; and the completely blurred “t, i” are deemed unrecognizable. This pixel-level fine-grained analysis frees the model from the chain hallucinations caused by whole-word recognition errors in traditional approaches.
>
> Second, the dynamic refusal strategy completely overturns the conventional output paradigm. When the system detects a blurry receipt, it no longer blindly outputs a guess as existing multimodal large language models (MLLMs) do, but instead opts to refuse to answer, thereby avoiding the generation of incorrect information.
>
> Finally, the precise guidance of the reward function, implemented via the GRPO algorithm, ingrains these behaviors as inherent model capabilities. As shown in Table 3, removing the refusal weight from the format reward immediately causes a 12% drop in the model’s performance on occluded character recognition.
>
> These three innovations together empower the model to achieve outstanding performance in complex degraded environments, significantly enhancing the reliability and accuracy of recognition.
>
> **Q9:**
>
>  Thank you for your comment. As shown in Figure 3, the format reward consists of two parts: the template format and the content (json) format. The purpose of the format reward is to encourage the model to follow a predefined template, facilitating the extraction of answers. For each generated result, the validation function checks whether the output adheres to the specified template format (i.e., "not clear enough Char-level OCR," "clear Char-level OCR," "Final OCR") as described in the Table1's caption. Additionally, it verifies whether the content complies with the standard, specifically whether it is in valid JSON format. We will provide more detailed descriptions in the main text in the future. Once again, thank you for your insightful comment.
>
> **Q10:**
>
> Thank you for your careful review and valuable feedback. We have thoroughly checked the manuscript and corrected all the typos and capitalization errors you pointed out (e.g., "MLMMs" to "MLLMs" at L106, "We" to "we" at L145, as well as the corrections for "KIE-HVQA" at L307 and "setting" at L320).

---

> > ### Comment · Reviewer_A3fL · 2025-08-03
> >
> > Thank you for the authors' rebuttal. The authors address some concerns, but only partially (approximately half).
> >
> > For Q1, did all data undergo human verification, or were only cases with differing evaluation results between GPT and Qwen further checked by humans? It is also confusing that you mention there are three types of predicted results: "Unrecognizable", "Partially Obsecured", "Visible". But this process only outputs binary "visible/invisible" results as labels, which do not correspond to the predicted results.
> >
> > For Q2, you only describe the label annotation process, but do not provide details on how accuracy is actually computed. Furthermore, in Section 3.3, the three criteria are referred to as "Legible Character Accuracy", "Degraded Character Accuracy", and "Global OCR Performance". However, in Table 1, the metrics are listed as "Clr", "Nc", and "Final". Are they the same metrics? If yes, why don't you use concistent naming?
> >
> > Q3 is well-explained, but I did not see a promise to revise the figure.
> >
> > For Q4, do you mean that you will add a new section detailing the SFT process?
> >
> > For Q5, you might miss my question. I am wondering how the metric is calculated. But you simply describe the functionality of each metric.
> >
> > For Q8, the first and second reasons you provided are understandable. However, the rationale for introducing GRPO remains unclear.

---

> ### Author Response · Authors · 2025-08-03
>
> Thank you for your comments. We hope that the following responses address the questions you raised. Here are our detailed replies to each comment:
>
> **Q1:**
>
> - We only conducted human verification on samples where the evaluation results of GPT and Qwen differed. When both models successfully recognize a degraded character, it is considered partially obscured but still recognizable, so no additional review is needed.
> - Our judgment process focuses solely on degraded characters, and therefore the labels are classified into two categories: “Degraded Visible” and “Degraded Not Visible.” Specifically, degraded characters are either “Degraded Visible” (partially obscured but still recognizable) or “Degraded Not Visible” (unrecognizable). While characters that have not undergone degradation are assumed to be fully visible and are not included in this labeling process. This is because the OCR datasets we use primarily focus on the task of recognizing “visible text,” so any characters without degradation can be regarded as “visible” characters. Thus, the “Degraded Visible/Degraded Not Visible” labels correspond to the previously mentioned types of “Partially Obscured” and “Unrecognizable,” simplifying the classification and avoiding confusion. We hope this explanation helps you better understand our labeling and evaluation procedure.
>
>
> **Q2:**
>
> - I will explain how accuracy is calculated. We compute the *edit distance* separately for characters with different occlusion levels and then calculate accuracy as: *1 - (edit_dist / max_len)*, where *max_len = max(len(pred), len(truth))*. This includes three parts: Legible Character Accuracy, Degraded Character Accuracy, and Global OCR Performance. The detailed formulas are in Algorithm 1.
>   For example, take the word "Beautiful" from Figure 4. We split it into legible characters (“B, a, u, f, u, l”), a partially occluded character (“e”), and the full string. Suppose the model predicts legible characters as “B, a, u, f, u, l”, partially occluded characters as “e, u”, and the full string as “Beauuful”.
>
>   1. For legible characters, prediction matches ground truth perfectly (edit distance 0), so accuracy is 100%.
>   2. For partially occluded characters, ground truth is “e” but prediction is “e, u” (edit distance 1, max length 2), so accuracy is 1 - 1/2 = 0.5.
>   3. For the full string, ground truth is “Beau ful”, prediction is “Beauuful” (edit distance 1, max length 8), resulting in accuracy 1 - 1/8 = 0.875.
>
> - For the second question, these metrics are indeed the same type of metric. These three indicators correspond exactly to the abbreviations in the table. We will make this clearer in subsequent revisions. Thank you for pointing this out.
>
>
> **Q3:**
>
> We will certainly revise and update the figures in the revised paper to more clearly and intuitively present the relevant content.
>
> **Q4:**
>
> We will add a new section that provides a detailed explanation of the cold-start SFT process to help readers better understand the details. Once again, thank you very much for your valuable suggestions and assistance!
>
> **Q5:**
>
> We apologize for the earlier misunderstanding. The metrics used in the reward function are consistent with the evaluation metrics to ensure that the model correctly interprets positive and negative feedback. Specifically, clear_metric, not_clear_metric, and final_metric correspond respectively to the Legible Character Accuracy, Degraded Character Accuracy, and Global OCR Performance described in Q2. All these calculations are based on edit distance. For detailed calculation methods, please refer to the explanation provided in Q2.
>
> **Q8:**
>
> The main reasons for introducing GRPO are as follows:
> 1. Enhanced reasoning ability: GRPO has been demonstrated to significantly improve model performance on reasoning tasks such as math and counting [1]. Since our design for character-level degradation recognition requires enhancing the model's reasoning capabilities, the GRPO method is a natural fit.
> 2. Improved generalization: Compared to SFT, which relies solely on supervised learning from labeled data, GRPO more effectively guides the model to adapt to unseen or differently distributed inputs, thereby significantly enhancing its generalization ability, an essential quality given the diverse and complex scenarios encountered in degraded text recognition.
> 3. Modeling dynamic abstention: SFT only learns the distribution of “correct” outputs from labeled data and cannot effectively capture patterns of “incorrect” generation. In contrast, GRPO enables the model to dynamically refuse to answer when faced with uncertain or erroneous generations, facilitating uncertainty modeling.
>
>
> [1] Tan, H., Ji, Y., Hao, X., Lin, M., Wang, P., Wang, Z., & Zhang, S. (2025). Reason-rft: Reinforcement fine-tuning for visual reasoning. arXiv preprint arXiv:2503.20752.

---

> > ### Comment · Reviewer_A3fL · 2025-08-06
> >
> > Most of my concerns are addressed. Therefore, I will raise my rating to borderline accept.

---

> > > ### Author Response · Authors · 2025-08-06
> > >
> > > Dear Reviewer A3fL,
> > >
> > > Thank you for your constructive feedback. We sincerely appreciate your time and effort in reviewing our manuscript. We are glad to hear that most of your concerns have been addressed, and we appreciate your updated rating.
> > >
> > > If there are any remaining issues or additional clarifications we can provide to further improve the manuscript, please let us know. We would be happy to supply further details or refinements as needed.
> > >
> > > Thank you once again for your valuable comments.
> > >
> > > Best regards,
> > > Authors of #16615

---

### Official Review · Reviewer_fNhj · 2025-07-02

**Clarity:** 2
**Significance:** 2
**Originality:** 2
**Rating:** 3
**Confidence:** 3

**Summary:**

This paper proposes the first benchmark specifically designed to evaluate OCR hallucination in degraded images, covering diverse document types such as identity cards, invoices, and prescriptions. To address this real-world challenge, the authors further introduce a Group Relative Policy Optimization (GRPO)-based training framework, incorporating a novel reward mechanism to fine-tune Multimodal Large Language Models (MLLMs). Experimental results demonstrate the effectiveness of the proposed framework on the newly introduced benchmark.

**Questions:**

Please refer to the listed weakness.

**Ethical Concerns:**

["NO or VERY MINOR ethics concerns only"]

**Final Justification:**

The rebuttal addressed some of my concerns, but I'm still not convinced regarding the technical contributions and model generalization toward other benchmarks.

**Limitations:**

Yes

**Paper Formatting Concerns:**

Null

**Quality:**

2

**Strengths And Weaknesses:**

Strength
1.	The paper proposes the OCR benchmark with degraded documents—a challenging yet realistic task setting that is crucial for enhancing the reliability of MLLMs.
2.	The proposed GRPO-based training framework effectively improves model performance on the newly introduced benchmark.

Weakness
1.	The benchmark lacks essential details. It would be better to provide comprehensive dataset documentation by including key statistics (e.g., size, category distribution) and a clear explanation of the dataset construction process. Additionally, incorporating human expert validation can improve the annotation quality and enhance the benchmark's overall credibility.
2.	As illustrated in Figure 2, the benchmark introduces numerous novel tokens (e.g., <part_occluded>) that are unseen during MLLM pretraining. Consequently, existing MLLMs are inherently incapable of generating these tokens. So the authors should provide more details about how to control comparison fairness between the trained model and the original model on generating unseen tokens.
3.	The proposed GRPO-based framework only modifies the reward design without introducing significant architectural or algorithmic innovations. So the contribution of this paper from a technical perspective should be strengthened.
4.	The method does not show improvement on traditional OCR benchmarks, which raises concerns that the observed gains on the proposed benchmark may result from overfitting to specific data patterns. So it would be better to validate the model’s effectiveness and generalization ability with broader experiments.

---

> ### Author Rebuttal · Authors · 2025-07-31
>
> **Q1:**
>
> We appreciate your attention to the details of the dataset. In Section 3.2, we provide a detailed description of the construction process of the KIE-HVQA benchmark: this dataset integrates 100 key information queries from OCRBench, entity data from WildReceipt, and 200 synthetic templates generated by GPT-4o (all using compliant fictional information), totaling 400 carefully designed test samples (see lines 129–156).
>
> Regarding degradation simulation, we applied pixel-level degradation to key regions using Python scripts and Photoshop, including various real-world degradation types such as motion blur, low contrast, and partial occlusion. Visual examples of these degradation effects can be found in Figure 2.
>
> Furthermore, to ensure annotation quality, we implemented a rigorous double-blind cross-validation mechanism — each data sample was independently reviewed by at least two domain experts. Additionally, we employed multi-model consistency checks to verify the reliability of the degradation region annotations. This layered quality assurance system significantly enhances the credibility of the benchmark.
>
> **Q2:**
>
> Our paper does not contain [object Object] tokens. You may be referring to the special tokens such as <part_occluded> and <occluded> shown in Figures 1 and 2. We clarify that these tokens are used solely as visual indicator and do not participate in the actual input or output processes of the models. To ensure fair evaluation, we designed a unified zero-shot evaluation prompt template (the full prompt is provided in the appendix), which effectively prevents evaluation bias caused by differences in model output formats.
>
> **Q3:**
>
> Our core innovation lies in task-driven refinement and integration. Traditional reinforcement learning methods typically require models to execute complex reasoning chains to obtain correct answers; by contrast, this paper innovatively unifies the GRPO method with refusal-to-answer tasks. Inspired by counting tasks, we introduce an indirect reasoning step: during the GRPO phase, the model first infers character quantities and then decides whether to refuse answering based on this inference. Building upon this mechanism, we propose the first hierarchical reward rules for character degradation, quantifying visual uncertainty into three trainable signals. Furthermore, through collaborative integration of GPT-4o and DeepSeek-R1, we construct vision-text reasoning chains that effectively overcome the limitations of pure text models in processing multimodal data.
>
> **Q4:**
>
> As shown in Table 2, our model maintains a comparable accuracy level to the original model on  OCR benchmarks (Scene: 180 vs. 181, Doc: 179 vs. 181), which clearly demonstrates that the optimization scheme does not compromise the fundamental OCR capability. It is important to emphasize that the core objective of KIE-HVQA is to enhance the model’s robustness against hallucinations in degraded scenarios, rather than directly optimizing traditional OCR metrics. On the degraded subsets of WildReceipt and OCRBench, the model achieves a breakthrough accuracy of 61.34% in recognizing blurred characters, strongly validating the approach’s generalization ability to real-world degradation patterns. Compared to GPT-4o and Qwen2.5-VL, this represents a relative improvement of +29.6% and +37.2%, respectively.

---

> > ### Comment · Reviewer_fNhj · 2025-08-06
> >
> > The rebuttal has addressed some of my concerns and I'll raise the rating. That said, I'm still not much convinced regarding the technical contributions and model generalization toward other benchmarks.

---

> > > ### Author Response · Authors · 2025-08-08
> > >
> > > Dear Reviewer fNhj,
> > >
> > > Thank you for your constructive feedback and for acknowledging our efforts in addressing some of your concerns. We sincerely appreciate your time and insights.
> > >
> > > Your guidance would greatly help us improve the paper, and we’re committed to addressing any concerns. Thank you again for your valuable comments.
> > >
> > > Best regards,
> > > Authors of #16615

---

### Official Review · Reviewer_AmPt · 2025-07-03

**Clarity:** 3
**Significance:** 3
**Originality:** 3
**Rating:** 4
**Confidence:** 3

**Summary:**

The paper presents KIE-HVQA, the first benchmark for measuring OCR hallucinations in degraded document images with character-level annotations. A 7B model is trained via supervised CoT followed by GRPO with a reward that penalizes hallucinations. The system achieves ~28% higher hallucination-free accuracy than GPT-4o on KIE-HVQA while matching OCR baselines on standard tasks, combining a novel benchmark and RL method for robust document understanding

**Questions:**

1. Have you tested the model on entirely unseen corporate or governmental documents to verify generalisation beyond the synthetic portion?
2. How does hallucination suppression change if degradation masks are obtained from an automated detector rather than ground truth?
3. Could the authors report results for a system that first runs a state-of-the-art OCR engine, then feeds text and images to the LLM? This would clarify the specific advantage of GRPO.

**Ethical Concerns:**

["NO or VERY MINOR ethics concerns only"]

**Final Justification:**

While your clarifications addressed many of my concerns, considering the overall quality, scope, I will keep my original score, which was already positive.

**Limitations:**

yes

**Quality:**

3

**Strengths And Weaknesses:**

**Strengths**
1. First, the benchmark itself fills a clear gap: existing OCR/VQA suites rarely test degraded inputs or explicitly label which characters are uncertain. By supplying pixel-level reliability masks and diverse degradation types, KIE-HVQA offers the community a concrete target for measuring and analysing hallucinations in safety-critical settings such as ID verification and medical forms.
2. Second, the technical solution is well motivated and experimentally validated. The GRPO framework, with its uncertainty-aware reward, yields large gains over strong proprietary and open-source baselines without eroding general OCR accuracy.

**Weaknesses**
1. Despite its realism, the dataset is modest in scale and partially synthetic, relying on GPT-4o-generated documents; it is unclear whether 2 400 samples cover the full variability of real-world paperwork or support robust generalisation to unseen layouts and languages.
2. The reward formulation assumes oracle-level degradation labels during training, practical deployment would lack such pixel-wise ground truth, so the paper should discuss how performance degrades when degradation detection itself is noisy or learned.
3. Finally, the evaluation omits specialised vision-only OCR engines or hybrid pipelines that could be combined with LLM reasoning; without these baselines it is hard to disentangle the benefit of the new learning paradigm from that of simply adding more OCR-centric supervision.

---

> ### Author Rebuttal · Authors · 2025-07-31
>
> **Q1:**
>
> Firstly, the training data in our work primarily originates from the TextOCR dataset, whereas the validation set includes other categories, such as cards, corporate documents, which were not encountered during training. Therefore, there exists a certain domain difference between the training and validation data. The experimental result of 58.05% demonstrates the generalizability of our method.
>
> Furthermore, we have conducted evaluations on the OCRBench dataset (as shown in Table 2), which further demonstrate the strong generalization capability of our method in conventional OCR scenarios.
>
> To improve the diversity of data types in the validation set, we plan to expand it in future work to enhance the comprehensiveness and representativeness of our evaluation.
>
> **Q2:**
>
> Our training process does not rely on bounding box annotations. The reason we use bounding boxes in the pipeline is primarily to generate accurate degradation samples, which in turn provide precise labels for training and enable controllable degradation through image processing techniques. Therefore, the entire process does not require assistance from automatic detectors. Our method adopts an end-to-end SFT-RL training framework that directly optimizes from pixel-level input to text output. In fact, if we can obtain a large amount of real training data with OCR annotations and accurate character-level degradation information, we do not depend on any coordinate-based detection tools, and the model performance is expected to improve further. We appreciate your valuable suggestion and will continue to explore the potential application of automatic detectors in this stage in our future work.
>
> **Q3:**
>
> Thank you for the valuable suggestion. We conducted experiments on a system that first runs a state-of-the-art OCR engine, then feeds the recognized text along with the images into the large language model. The results are shown in the table below:
>
> | Dataset   | Clr | Nc | Final |
> |-----------|----------------------|-------------------------------|--------------------------|
> | Card      | 14.11                | 0.0                           | 15.13                    |
> | OCRBench  | 38.85                | 0.0                             | 42.50                    |
> | Wild      | 5.25                 | 0.0                           | 4.85                     |
>
> As can be seen, the model blindly trusts the OCR outputs in the OCR+LLM system, leading to a significant drop in recognition performance on unclear or blurred text.

---

> ### Author Response · Authors · 2025-08-06
>
> Dear Reviewer AmPt,
>
> As the discussion period deadline approaches, I would like to express our sincere gratitude for your time and valuable feedback on our paper. We have carefully addressed all comments in our previous rebuttal and hope our responses have adequately clarified your concerns.
>
> Please let us know if any additional questions or clarifications are needed. We would be more than happy to provide further details. Your comments have been critical in improving this work, and we deeply appreciate your efforts.
>
> Looking forward to your guidance. Thank you again for your consideration.
>
> Best regards,
>
> Authors of #16615

---

> ### Comment · Reviewer_AmPt · 2025-08-09
>
> Thank you for the detailed response. While your clarifications addressed many of my concerns, considering the overall quality, scope, I will keep my original score, which was already positive. Wishing the authors the best of luck.

---

### Official Review · Reviewer_cy4C · 2025-07-03

**Clarity:** 3
**Significance:** 3
**Originality:** 3
**Rating:** 4
**Confidence:** 4

**Summary:**

This paper introduces KIE-HVQA, the first benchmark explicitly designed to evaluate OCR hallucinations in multimodal large language models (MLLMs) under degraded visual conditions. The benchmark includes degraded document images with pixel-level OCR reliability annotations, enabling precise assessment of hallucination behavior. To address this challenge, the authors propose a novel reinforcement learning framework using Group Relative Policy Optimization (GRPO), combined with a new reward function that accounts for visual uncertainty. The approach incorporates both supervised fine-tuning and reinforcement learning to encourage models to recognize and appropriately respond to degraded visual information. Experiments on Qwen2.5-VL show substantial improvements over state-of-the-art models like GPT-4o, especially in hallucination-prone settings, while preserving general OCR performance.

**Questions:**

Prior works[3] have tried to address a similar problem of models not focusing on visual details. Under settings like TextVQA[1], STVQA[2], have the authors done any experiments or have any thoughts as to how the authors' proposed method would work?

[1] Singh, A., Natarajan, V., Shah, M., Jiang, Y., Chen, X., Batra, D., Parikh, D. and Rohrbach, M., 2019. Towards vqa models that can read. In Proceedings of the IEEE/CVF conference on computer vision and pattern recognition (pp. 8317-8326). \
[2] Biten, A.F., Tito, R., Mafla, A., Gomez, L., Rusinol, M., Valveny, E., Jawahar, C.V. and Karatzas, D., 2019. Scene text visual question answering. In Proceedings of the IEEE/CVF international conference on computer vision (pp. 4291-4301). \
[3] Hegde, S., Jahagirdar, S. and Gangisetty, S., 2023. Making the v in text-VQA matter. In Proceedings of the IEEE/CVF Conference on Computer Vision and Pattern Recognition (pp. 5580-5588).

**Ethical Concerns:**

["NO or VERY MINOR ethics concerns only"]

**Final Justification:**

The authors have addressed my initial concerns satisfactorily. The paper is technically sound and the contributions are clear, I believe the overall impact, novelty, and scope remain somewhat limited to OCR recognition. I believe the work would be much more effective when aligned with reasoning rather than just recognition which is although important, would be better with certain revisions. Therefore, I am maintaining my rating.

**Limitations:**

Yes

**Quality:**

3

**Strengths And Weaknesses:**

**Strengths**
- The paper presents a rigorous and thorough experimental setup, including well-designed benchmarks, precise annotation strategies, and a custom reward function tailored to OCR degradation. The results clearly demonstrate strong gains in hallucination suppression and robustness across diverse degradation types.
- This work tackles a critical and underexplored problem in OCR image understanding, vision-language hallucination in degraded inputs. It offers not only a novel dataset (KIE-HVQA) but also a method to mitigate OCR errors.
- The proposed GRPO-based framework, with a multi-objective reward function, is a novel approach to mitigating OCR hallucinations. The emphasis on "vision-faithful reasoning" and incorporating uncertainty-driven refusal behaviors directly into the training process via reinforcement learning is innovative and well-aligned.

**Weakness**
- While the paper effectively demonstrates the problem and its proposed solution, it lacks a comprehensive comparison with other potential or existing mitigation techniques for hallucination in MLLMs (beyond just showing that other models hallucinate). A discussion or comparison with methods that might approach hallucination from different angles (ex., uncertainty quantification, confidence estimation, or alternative training paradigms) would provide a broader context and further highlight the unique advantages of the proposed GRPO-based approach.
- The reinforcement learning approach is adapted from prior work (ex., DeepSeek-R1), and while the integration with OCR-specific reward design is well-executed, it builds incrementally on known RL optimization principles.
- Lack of comparison and evaluation on other widely used datasets like TextVQA[1], ST-VQA[2], OCR-VQA[3]. These datasets contain OCR images and bounding box data along with them.

[1] Singh, A., Natarajan, V., Shah, M., Jiang, Y., Chen, X., Batra, D., Parikh, D. and Rohrbach, M., 2019. Towards vqa models that can read. In Proceedings of the IEEE/CVF conference on computer vision and pattern recognition (pp. 8317-8326). \
[2] Biten, A.F., Tito, R., Mafla, A., Gomez, L., Rusinol, M., Valveny, E., Jawahar, C.V. and Karatzas, D., 2019. Scene text visual question answering. In Proceedings of the IEEE/CVF international conference on computer vision (pp. 4291-4301). \
[3] Mishra, A., Shekhar, S., Singh, A. K., and Chakraborty, A., "OCR-VQA: Visual Question Answering by Reading Text in Images," 2019 International Conference on Document Analysis and Recognition (ICDAR), pp. 947-952

---

> ### Author Rebuttal · Authors · 2025-07-31
>
> **Q1:**
>
> Thank you for your valuable suggestion. Current mainstream research on multimodal hallucinations[1] mostly focuses on objects, attributes, or relationships, whereas our paper concentrates on OCR hallucinations in degraded documents, which can be viewed as an exploration of existence and attribute hallucinations for text—a special category of objects. Our method essentially constitutes an uncertainty quantification modeling approach, introducing a rejection mechanism in degraded scenarios that aligns with the underlying ideas of most approaches addressing existence hallucinations. What differentiates our work is the use of natural language descriptions such as "occluded" and "invisible" to measure uncertainty, combined with reinforcement learning modeling, thereby providing a clearer solution tailored to hallucination-prone scenarios specific to OCR tasks. We will incorporate comparisons with traditional hallucination tasks in the related work section.
>
> **Q2:**
>
> Our core innovation lies in task-driven refinement and integration. Traditional reinforcement learning methods typically require models to perform complex chains of reasoning to arrive at correct answers. In contrast, this paper innovatively unifies the GRPO method with refusal-to-answer tasks and, for the first time, proposes a quantifiable dataset addressing the hallucination problem in degraded text OCR, providing a foundation for future research.
>
> Inspired by counting tasks[2], we introduce an indirect reasoning step: during the GRPO phase, the model first infers character quantities and then decides whether to refuse answering based on this inference. Building upon this mechanism, we propose the first hierarchical reward rules for character degradation, quantifying visual uncertainty into three trainable signals. Based on experimental results, our proposed method, which constructs vision-text reasoning chains through the collaborative integration of GPT-4o and DeepSeek-R1, successfully alleviates the hallucination problem in degraded text OCR.
>
> **Q3:**
>
> This paper primarily evaluates the model’s general OCR capability (as shown in Table 2, demonstrating comparable results to leading models). It should be noted that although these datasets contain OCR images and bounding box information and are widely used to train MLLMs, they consist almost entirely of clear and easily readable text, with little to no degraded text samples. As a result, they are insufficient for assessing model performance on degraded OCR inputs, which is the focus of our evaluation. Therefore, our evaluation framework cannot be directly applied to these datasets. As future work, we plan to construct more datasets by combining degradation techniques with large model annotations and manual verification, and will release these publicly to better support the evaluation methodology proposed in this paper.
>
> **Q4:**
>
> We sincerely thank the reviewer for the careful attention to the work "Making the V in Text-VQA Matter." This prior work indeed makes important contributions toward addressing the issue of models overly relying on textual cues and neglecting visual details in TextVQA tasks. By employing a multimodal embedding fusion framework, it maps image objects, OCR texts, and question texts into a unified representation space, effectively balancing the visual and linguistic modalities.
> In contrast, our work focuses on a fundamentally different and challenging problem:
> - Firstly, we mainly focus on the hallucination problem caused by the model’s over-reliance on large language model (LLM) prior knowledge in degraded scenarios (such as low-quality images), which is different from previous works that address the issue of weak attention to text within images.
> - Secondly, we propose a character-level fine-grained alignment mechanism, which differs from the coarser multimodal fusion approaches in prior work, aiming to capture textual details more precisely;
> - Lastly, we design an end-to-end architecture that directly learns text representations from pixels **without relying on external OCR systems**, thereby avoiding additional error propagation and improving overall robustness.
> Therefore, while both works share a common interest in vision-language fusion, our research emphasizes distinct challenges and adopts different technical approaches. We hope this explanation clarifies the differences in focus and contributions between the two works. We greatly appreciate the reviewer’s insightful comments and valuable feedback.
>
> [1] Bai, Z., Wang, P., Xiao, T., He, T., Han, Z., Zhang, Z., & Shou, M. Z. (2024). Hallucination of multimodal large language models: A survey. arXiv preprint arXiv:2404.18930.
>
> [2] Tan, H., Ji, Y., Hao, X., Lin, M., Wang, P., Wang, Z., & Zhang, S. (2025). Reason-rft: Reinforcement fine-tuning for visual reasoning. arXiv preprint arXiv:2503.20752.

---

> > ### Comment · Reviewer_cy4C · 2025-08-06
> > **Official Comment by Reviewer cy4C**
> >
> > I thank the authors for their thorough rebuttal. They have addressed my initial concerns satisfactorily, and I appreciate the clarifications provided.
> >
> > That said, while the paper is technically sound and the contributions are clear, I believe the overall impact, novelty and scope remain somewhat limited to OCR recognition. Therefore, I am maintaining my rating.

---

> > > ### Author Response · Authors · 2025-08-08
> > >
> > > Dear reviewer cy4C,
> > >
> > > We are pleased to hear that **your concerns have been fully addressed**, and we appreciate your recognition that our work is **technically sound and the contributions are clear**. Your constructive comments are crucial to our work, and we thank you again for your help in improving it. As the discussion phase is approaching the deadline, if you have any other concerns that require further clarification, we are happy to respond to you promptly.
> > >
> > > Best regards,
> > >
> > > Authors of #16615

---

### Note · Authors · 2025-08-14

Dear Chairs and Reviewers,

We are grateful that all reviewers acknowledged our rebuttal addressed most of their concerns. We appreciate reviewers A3fL and fNhj for **raising** their ratings, and reviewers cy4C and AmPt for **maintaining their positive** ratings.

Here's a concise remark of how we addressed key concerns:

1. [cy4C] Comparison concerns: Clarified our unique focus on text-specific hallucinations with uncertainty-aware refusal mechanisms, explaining why existing datasets are insufficient for our task. The reviewer acknowledged our response, confirming the technical soundness of our work.

2. [AmPt] Generalizability questions: Demonstrated cross-domain effectiveness through disjoint training/validation data, showed our end-to-end approach avoids reliance on degradation detectors, and provided evidence that OCR+LLM pipelines fail on degraded text. The reviewer recognized the validity of our explanations.

3. [A3fL] Methodology details: Detailed our dual-model validation approach (GPT-4o + Qwen2.5-VL) for annotations, clarified metrics, and improved descriptions of our SFT-GRPO pipeline. The reviewer raised their rating after our clarifications.

4. [fNhj] Technical contribution: We believe our work makes meaningful contributions: focusing on the **under-explored but critical problem** of OCR hallucination, filling a critical gap with the **first OCR hallucination benchmark** for degraded documents, and proposing an **effective GRPO-based solution** that improves hallucination-free accuracy by ~28% over GPT-4o on KIE-HVQA without sacrificing standard OCR performance on OCRBench. We thank the reviewers for considering our technical contributions as "clear" (Reviewer cy4C), "valuable" (Reviewer A3fL), and "well motivated and experimentally validated" (Reviewer AmPt).

We sincerely appreciate the valuable time of all reviewers, ACs, SACs and PCs. We commit to incorporating all discussion feedback into the final paper and appreciate your consideration.

Sincerely,

The Authors of Submission #16615

---

### Decision · Program_Chairs · 2025-09-17

**Decision:**

Accept (poster)

**Comment:**

The paper focuses on hallucinations for MLLM in OCR.
They also propose a RL-based approach with an heuristic reward to improve or mitigate hallucinations.

From the reviews, the paper is technically sound with clear contributions. A major limitation is the scope, being focused on OCR recognition. While agreeing that OCR might be limited, the novelty of the approach and tackling this relevant problem might inspire other approaches in other domains. Therefore leaning towards acceptance.

Importantly, reviewers acknowledge the efforts made to address all the comments